# Arachidonic Acid as Mechanotransducer of Renin Cell Baroreceptor

**DOI:** 10.3390/nu14040749

**Published:** 2022-02-10

**Authors:** Undurti N. Das

**Affiliations:** UND Life Sciences, 2221 NW 5th St., Battle Ground, WA 98604, USA; undurti@lipidworld.com

**Keywords:** renin, arachidonic acid, juxtaglomerular cells, granular cells, nucleus, lamin, actin-myosin, afferent

## Abstract

For normal maintenance of blood pressure and blood volume a well-balanced renin-angiotensin-aldosterone system (RAS) is necessary. For this purpose, renin is secreted as the situation demands by the juxtaglomerular cells (also called as granular cells) that are in the walls of the afferent arterioles. Juxtaglomerular cells can sense minute changes in the blood pressure and blood volume and accordingly synthesize, store, and secrete appropriate amounts of renin. Thus, when the blood pressure and blood volume are decreased JGA cells synthesize and secrete higher amounts of renin and when the blood pressure and blood volume is increased the synthesis and secretion of renin is decreased such that homeostasis is restored. To decipher this important function, JGA cells (renin cells) need to sense and transmit the extracellular physical forces to their chromatin to control renin gene expression for appropriate renin synthesis. The changes in perfusion pressure are sensed by Integrin β1 that is transmitted to the renin cell’s nucleus via lamin A/C that produces changes in the architecture of the chromatin. This results in an alteration (either increase or decrease) in renin gene expression. Cell membrane is situated in an unique location since all stimuli need to be transmitted to the cell nucleus and messages from the DNA to the cell external environment can be conveyed only through it. This implies that cell membrane structure and integrity is essential for all cellular functions. Cell membrane is composed to proteins and lipids. The lipid components of the cell membrane regulate its (cell membrane) fluidity and the way the messages are transmitted between the cell and its environment. Of all the lipids present in the membrane, arachidonic acid (AA) forms an important constituent. In response to pressure and other stimuli, cellular and nuclear shape changes occur that render nucleus to act as an elastic mechanotransducer that produces not only changes in cell shape but also in its dynamic behavior. Cell shape changes in response to external pressure(s) result(s) in the activation of cPLA2 (cytosolic phospholipase 2)-AA pathway that stretches to recruit myosin II which produces actin-myosin cytoskeleton contractility. Released AA can undergo peroxidation and peroxidized AA binds to DNA to regulate the expression of several genes. Alterations in the perfusion pressure in the afferent arterioles produces parallel changes in the renin cell membrane leading to changes in renin release. AA and its metabolic products regulate not only the release of renin but also changes in the vanilloid type 1 (TRPV1) expression in renal sensory nerves. Thus, AA and its metabolites function as intermediate/mediator molecules in transducing changes in perfusion and mechanical pressures that involves nuclear mechanotransduction mechanism. This mechanotransducer function of AA has relevance to the synthesis and release of insulin, neurotransmitters, and other soluble mediators release by specialized and non-specialized cells. Thus, AA plays a critical role in diseases such as diabetes mellitus, hypertension, atherosclerosis, coronary heart disease, sepsis, lupus, rheumatoid arthritis, and cancer.

## 1. Introduction

Maintenance of normal blood pressure and blood volume is essential for health. Renin-angiotensin-aldosterone system (RAS) pays a crucial role in this aspect. Renin is a crucial part of the RAS. Juxtaglomerular (JG) cells also called as granular cells (renin cells) are the seat of renin synthesis and they store and secrete renin as the situation demands (see Figure 1). JG cells are in the walls of the afferent arterioles of the glomerulus and are considered as the specialized smooth muscle cells. In view of the tight packing of the cells of the juxtaglomerular apparatus especially of the granular cells between the endothelial cells, myocytes of the afferent arteriole and other cells (see Figure 1) even slight changes in the blood volume and pressure within the afferent arteriole are sensed by the renin cells (granular cells). Thus, the renin expressing cells are uniquely located to sense and respond to changes in blood pressure and the extracellular fluid such that they can either enhance or decrease the synthesis and release of renin to restore homeostasis. Recent studies revealed that the baroreceptor function of the renin cells can be attributed to its nuclear mechanotransducer ability to sense and transmit the extracellular physical forces to its chromatin to regulate renin gene expression and its secretion. Perfusion pressure changes sensed by the integrin β1 are transmitted to the renin cell’s nucleus via lamin A/C that results in changes in the architecture of the chromatin and corresponding alterations in the expression of the renin gene [1]. It is likely that the changes in the perfusion pressure sensed by the renin cells and the renin gene via a nuclear mechanotransduction mechanism is regulated or controlled by certain intermediate molecules. I propose that such an intermediate molecule(s) could be arachidonic acid (AA) and its metabolites that have modulatory influence on the secretion and action of RAS.

## 2. JGA (Juxta Glomerular Apparatus) Functions as a Baroreceptor

Baroreceptors are mechanoreceptor sensory neurons capable of responding to changes in the stretch of the blood vessel. Thus, a change in the pressure of blood vessel triggers an alteration in the action potential generation rates that is conveyed to the solitary nucleus in the medulla oblongata via autonomic reflexes to influence the cardiac output and vascular resistance [2].

Baroreceptors are of two types: high-pressure arterial baroreceptors and low-pressure baroreceptors (also known as cardiopulmonary or volume receptors) which are located within the carotid sinuses and the aortic arch (high pressure baroreceptors) and within the atria, ventricles, and pulmonary vasculature (low-pressure baroreceptor) respectively.

The main function of the baroreceptors is to maintain systemic blood pressure. Increase in blood pressure stretches the baroreceptors resulting in an increase in the vagal tone nerve and inhibition of the sympathetic outflow that results in restoration of blood pressure to normal. In contrast, fall in blood pressure decreases signal output that disinhibits the sympathetic control sites and decreases parasympathetic activity resulting in an increase in blood pressure. Stimulation of the low-pressure baroreceptors enhances salt and water retention in addition to influencing intake of salt and water.

The information carried from the baroreceptors is integrated in the medulla oblongata that communicates with the heart muscle, the cardiac pacemaker and the arterioles and veins of the body. Any reduction in the sympathetic innervation of the kidneys causes vasodilation and increased blood flow to the kidneys resulting in an increase in urine production. On the other hand, whenever blood pressure falls, the sympathetic nerves stimulate renin release that ultimately results in the production of angiotensin II, which, in turn, enhances aldosterone that increases retention of Na^+^ and water. It is interesting that changes in afferent arterial pressure alters glomerular filtration rate.

It is noteworthy that kidneys regulate the rate of blood flow over a wide range of blood pressures. Despite significant changes in the blood pressure, the glomerular filtration rate changes very little due to two internal autoregulatory mechanisms: *the myogenic mechanism* and the *tubuloglomerular feedback mechanism* that operate independent of the outside influence.

## 3. Arteriole Myogenic Mechanism

When blood pressure increases, smooth muscle cells located in the arteriole wall get stretched resulting in its contraction to resist the increase in the blood pressure. As a result, very little change in the flow occurs. In contrast, when blood pressure is decreased, the smooth muscle cells relax leading to a decrease in the resistance that allows a continued even flow of blood.

## 4. Tubuloglomerular Feedback

JGA is involved in the tubuloglomerular feedback mechanism (Figure 1) that utilizes adenosine triphosphate (ATP), adenosine, and nitric oxide (NO) that are capable of either contract or relax the afferent arteriolar smooth muscle cells. The macula dense cells that are in intimate contact with the afferent and efferent arterioles of the glomerulus, respond to changes in the fluid flow rate and Na^+^ concentration. Activated macula densa cells release ATP and adenosine that stimulate the myogenic juxtaglomerular cells of the afferent arteriole to slow blood flow and reduce glomerular filtration rate. On the other hand, decrease in glomerular filtration rate results in less Na^+^ excretion that results in decreased ATP and adenosine production leading to the afferent arteriolar dilatation and increase in the glomerular filtratation rate. NO relaxes the afferent arteriole whereas ATP and adenosine constrict it. Thus, a close interaction exists between NO and adenosine and ATP on the glomerular filtratation rate.

The distal convoluted tubule that is in direct contact with the arterioles forms a part of the JGA known as the macula densa, which can monitor the fluid composition that is flowing through the distal convoluted tubule. Thus, any changes in the concentration of Na^+^ and the rate of fluid movement in the tubule macula densa leads to releases ATP and adenosine that regulate the glomerular filtration rate.

The regulation of renin release is done by the macula densa cells of the JGA of the afferent arteriole (Figure 1 and Figure 2). Renin, in turn, modulates the formation of angiotensin II, a stimulator of the release of aldosterone from the adrenal cortex. Aldosterone enhances Na^+^ reabsorption by the kidney and along with which water retention occurs leading to an increase in blood pressure.

## 5. Mechanotransduction from Cell Membrane to the Nucleus

Cell membrane integrity is essential for optimal cell response to various stimuli (both external and internal) since, all stimuli need to be conveyed to the genome through the cell membrane and vice versa [3]. This applies to the cells of the JGA and their response to various stimuli that regulate renin secretion. It has been reported that the nucleus serves as an elastic mechanotransducer of cellular shape deformation and controls its (cell) dynamic behavior [4,5]. Changes in the cell shape in response to external pressures induce inner nuclear membrane unfolding that, in turn, activates cPLA2 (cytosolic phospholipase 2) leading to myosin II recruitment resulting in actin-myosin cytoskeleton contractility (see Figure 3 and Figure 4). These events produce alterations in the perfusion pressure in the afferent arterioles with corresponding changes in the renin cell membrane and parallel changes in renin release [2]. Thus, renin cells that are near the afferent arteriolar endothelial cells/myocytes/mesangium glomerular cells sense changes in the afferent arterioles and convey the same to the renin cell membrane. This results in the activation of cPLA2 (and possibly other phospholipases).

Cell shape changes occur in response to pressure and stretch stimuli. This causes the nuclear membrane unfolding and activates cPLA2 resulting in release of AA (and other unsaturated fatty acids) from the cell membrane. Released AA induces changes in the actin cytoskeleton behavior and other cytoskeleton structures (including lamin, integrin β1). In response to changes in the mechanical forces (pressure and stretch) the cells (especially myocytes, macular densa, granular cells, mesangium-extra-glomerular cells, afferent and efferent endothelial cells) respond and interact with each other and transduce mechanical signaling through focal adhesion contacts and F-actin and other cytoskeleton structures (both extra and intracellular) that are transmitted to the nucleus through the nuclear pores and nuclear translocation of mechanosensing-dependent transcription factors (YAP-TAZ and MRTF- transcription factors). This leads to chromatin activation with consequent changes in the expression of renin gene. If tumor cells, leukocytes, or macrophages, etc., are exposed to pressure and stretch stimuli, cells migrate or metastasize. AA (and other unsaturated fatty acids) released lead to the formation of its metabolites that produce changes in the cell size, shape, alter motility, phagocytosis, and influence inflammation, immune response and show microbicidal action (see Figure 5).

Perfusion pressure changes and/or direct mechanical stimuli produce significant changes in renin gene expression and its phenotype. This has been attributed to the baroreceptor(s) that reside in renin cells that are sensed by the integrin β1 at the renin cell membrane. The stimuli that are sensed by the baroreceptors of the renin cell are transduced to the nuclear membrane and chromatin by lamin A/C that changes the renin gene expression and renin bioavailability such that homeostasis is restored [1,2]. These results imply that changes in the cell membrane dynamics are transmitted to the nucleus for changes in the respective genes. This transmission of messages from the cell membrane to the nucleus is performed by the cytoskeleton system that function as the second messenger of the mechanotransduction process. Perfusion pressure changes are conveyed to the renin cell membrane such that alterations in cPLA2 activity occurs in afferent arteriolar endothelial cells/myocytes/mesangium glomerular cells and renin cells.

## 6. AA Functions as a Mechanotransducer

Dietary essential fatty acids (EFAs) cis-linoleic acid (LA, 18:2 n-6) and alpha-linolenic acid (ALA, 18:3 n-3) are essential for humans to survive. EFAs are converted to their long-chain metabolites gamma-linolenic acid (GLA, 18:3 n-6), dihomo-GLA (DGLA, 20:3 n-6) and arachidonic acid (AA, 20:4 n-6) derived from LA and eicosapentaenoic acid (EPA, 20:5 n-3) and docosahexaenoic acid (DHA, 22:6 n-3) from ALA. All these fatty acids form an important constituent of cell membrane and are in their (cell membrane) phospholipid fraction. PUFAs by virtue of their unsaturation can enhance the fluidity of the cell membranes (including nuclear and mitochondrial membranes). Thus, it is envisaged that increased presence of PUFAs (especially AA, EPA and DHA) enhances the fluidity of the cell membrane whereas cholesterol and other saturated fatty acids render membrane more rigid. The importance of membrane fluidity lies in the fact that increased cell membrane fluidity increases the number of receptors and their affinity to their respective proteins/growth factors and hormones (reviewed in [3]). This is especially relevant to insulin action and development of insulin resistance. The cell membrane fluidity is also an important factor to enable cells such as leukocytes, macrophages, platelets, T and B cells and tumor cells to adhere to the extracellular matrix and other tissues and enable them to pass through capillaries. In the event the cell membrane of is sufficiently fluid, they will be able to pass through the capillaries some of which are much smaller in diameter compared to various blood cells and prevent thrombosis whereas, in general, tumor cells are sufficiently rigid that is responsible for their ability to produce vascular thrombosis.

The ability of cells to respond to various pressures and stretch stimuli (including shear stress of blood flow) results in cell membrane and nucleus to act as elastic mechanotransducer of cellular shape deformation and control their (cell) dynamic behavior including change in their shape, motility, ability to secrete chemicals needed to produce their respective actions, phagocytosis (endocytosis and exocytosis), inflammation, immune response, and other functions. In this context, the ability of AA (and DGLA, EPA and DHA) to form precursors to various biologically active metabolites assumes significance. The observation that nucleus acts as an elastic mechanotransducer of cellular shape deformation that results in the activation of cPLA2 and consequent release of AA (and possibly, DGLA, EPA and DHA; [4,5]) implies that the released fatty acids could be utilized for the formation of various metabolites from these released fatty acids (see Figure 5). Thus, it is envisaged that the released AA (and DGLA, EPA, and DHA) is converted into various metabolites that have several biological actions depending on the necessity. In this context, it is noteworthy that various metabolites formed from AA (prostaglandins, leukotrienes, thromboxanes, lipoxins; and those formed from EPA and DHA including resolvins, protectins and maresins) have very short half-life (few seconds) but potent biological actions that ensures much-needed local actions that are short lived but sufficiently strong to meet the local needs.

**Figure 5 nutrients-14-00749-f005:**
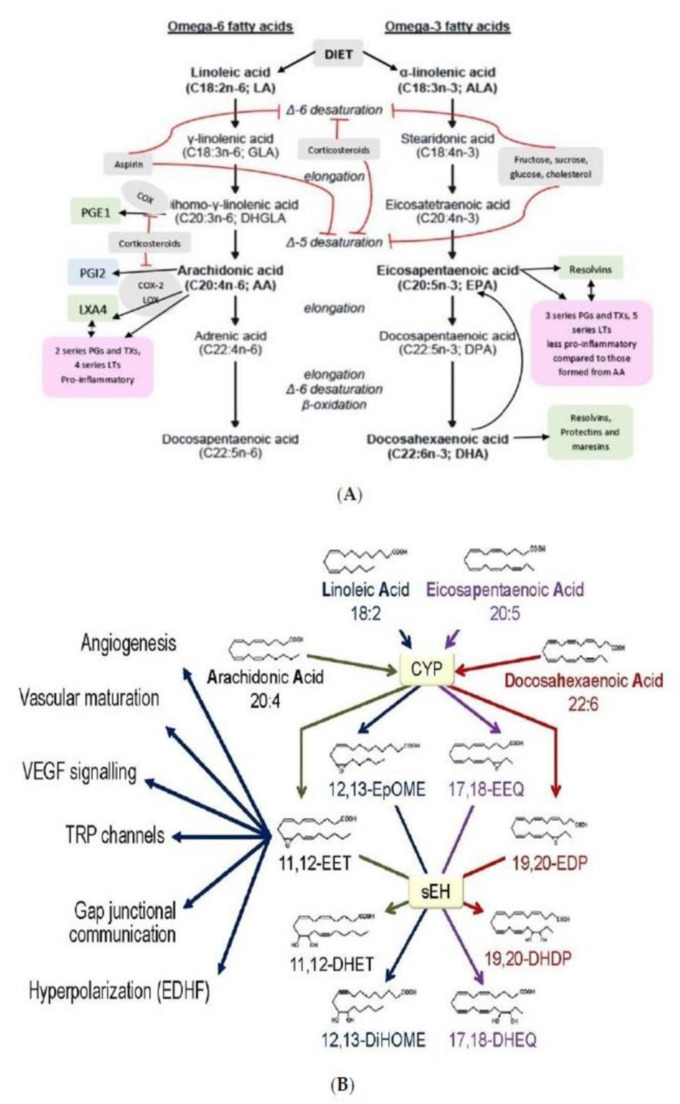
Scheme showing the metabolism of EFAs, and the various products formed from GLA, DGLA, AA, EPA, DPA and DHA. CYP = Cytochrome P450 enzymes sHE = Soluble epoxy hydrolase. (**A**) Metabolism of EFAs LA and ALA. (**B**) Metabolism of LA and EPA by cytochrome P450 enzymes and their actions. (**C**) Metabolism of DGLA and the products from the same. (**D**) Metabolism of AA and formation of LXA4 from the same. (**E**) A comparison of the metabolism of DGLA and AA and various products formed from these fatty acids. (**F**) A comparison of the metabolites formed from AA, DPA, EPA and DHA. (**G**) Metabolism of PUFAs and products formed from the same by the action of various enzymes.

## 7. AA and Its Metabolites Regulate Gene(s) Expression and Renin Release and Action

AA and its products regulate the release of renin at least, in part by producing changes in the transient receptor potential vanilloid type 1 (TRPV1) expressed in renal sensory nerves [6,7,8,9,10]. This implies that AA and its metabolites function as mediators that bring changes in renin gene expression in response to alterations in perfusion pressure involving nuclear mechanotransduction mechanism.

Several studies revealed that PGE2, PGI2, PGE synthase and PGE2 receptors EP2 and EP4 regulate renin secretion by the macula densa [11,12]. All four PGE2 receptor subtypes (EP1, EP2, EP3, EP4) control renal vascular tone, EP1 and EP3 receptors increasing, and EP2 and EP4 receptors, decreasing it [13]. PGE2 produces both renal vascular dilatation at low concentrations (1 nmol/) and vasoconstriction at higher concentrations (100 nmol/) [13].

This dual action of PGE2 (formed from AA) depends on the expression of PGE2 receptors, their number, and the concentration of PGE2 formed locally that, in turn, depends on AA released from the afferent arteriolar endothelial cells/myocytes/mesangium glomerular cells. The presence of EP1, EP2, EP3 and EP4 receptors in the vessels as well as in the thick ascending limb and collecting duct, glomerulus and collecting duct and the ability of PGE2 to modulate vascular tone and epithelial transport suggests that it regulates renin release, blood pressure and blood volume in response to various stimuli [14].

Akin to the actions of PGE2 on vascular tissue (both as a vasodilator and vasoconstrictor), it also has both pro- and anti-inflammatory actions. The pro-inflammatory actions of PGE2 can be ascribed to its ability to polarize macrophages by acting on mesenchymal stem cells (MSCs) binding to its receptors EP2 and EP4 depending on its local concentration [15,16,17,18,19,20,21,22]. Despite the belief that PGE2 is pro-inflammatory in nature, several studies suggested that PGE2 can also trigger anti-inflammatory events [17,18,19,20,21,22,23,24,25]. The pro- and anti-inflammatory actions of PGE2 depend on its ability to bind to different types of PGE2 receptors (EPs). At low concentrations, PGE2 binds to the high affinity EP4 receptor to enhance IL-23 production, whereas high PGE2 concentrations bind to EP2 receptor to suppress IL-23 release [19,20] that explains its dual pro and anti-inflammatory actions [26,27,28,29]. In addition, once PGE2 concentrations reach a peak, it triggers the generation of LXA4 and suppresses LTB4 production by inducing 5- and 15-lipoxygenases that triggers the pro-inflammatory status to be switched over to anti-inflammatory pathway (see Figure 6, [22,30]). These observations [15,16,17,18,19,20,21,22,23,24,25,26,27,28,29,30,31] emphasize the critical role of PGE2 in inflammation, vasoconstriction, vasodilatation and in the regulation of renin release from the JGA. This switchover of the metabolism of AA from PGE2 to LXA4 depends on the source of AA. There is a biphasic release of AA from the cell membrane lipid pool. The first pulse of AA release due to the activation of iPLA2 is utilized to form predominantly PGE2. On the other hand, the second pulse of AA release due to cPLA2 and/or sPLA2 activation is directed to form LXA4. Thus, factors regulating the activation of different forms of PLA2 are critical in the regulation of inflammation, vasoconstriction or vasodilatation and release or inhibition of renin release. It is likely that PGE1 derived from DGLA, the precursor of AA, also has actions like PGE2 in triggering the formation of LXA4 (see Figure 6) though PGE1 may be much less potent compared to PGE2. EPA and DHA also regulate inflammation, vascular tissue response and renin release. PGE3 and LTs (formed from EPA) and resolvins, protectins and maresins (derived from EPA and DHA) regulate inflammation, vascular tone, and renin release like PGE2, LTE4 and LXA4 (derived from AA) except that they are much less potent compared to those derived from AA.

The critical role of AA in various cellular processes including inflammation, vascular tone and renin release is supported by the observation that its peroxidized products bind to DNA and regulate gene(s) expression(s) [32,33]. It is likely that AA that is released in response to the changes in pressure and stretch get converted into its various metabolites (PGs, LTs, TXs, and LXA4) to regulate renin release. The possibility that AA by itself can regulate renin release cannot be discounted.

## 8. AA and Its Metabolites in Renin Synthesis, Secretion, and Action

Renin is an aspartyl protease which limits the activity of renin-angiotensin-aldosterone system (RAS). Renin-expressing cells in the kidney synthesize active renin from prorenin, which is stored in the cells to release it on demand. Both cAMP and Ca^2+^ pathways, cAMP-binding protein (CREB), and PPAR-γ (peroxisome proliferator-activated receptor-γ) exert a positive effect on renin gene transcription. In contrast to this, transcription factor NF-κB (nuclear factor-κB) interferes with the CREB-binding site and thus, inhibits the pro-inflammatory cytokines action on renin gene expression [34,35]. This may explain the negative feed-back regulation exerted by angiotensin-II/aldosterone on renin gene expression (and thus reduce renin secretion and levels). In contrast to this, AA, PGE1, PGE2, LXA4, resolvins, protectins and maresins may enhance renin gene expression by virtue of their inhibitory action on NF-κB, IL-6 and TNF-α, whereas AA metabolites, 12(s)-HPETE and 12-HETE, which are pro-inflammatory molecules inhibit renin gene expression [3,6,8,9,11,29]. These results emphasize as to how AA can both enhance and reduce renin synthesis and release and regulate blood volume and blood pressure. Based on these results, it is envisaged that AA release that occurs in response to changes in the pressure and stretch stimuli regulates renin release. Whenever there is a decrease in pressure and stretch stimuli as a result reduced blood volume and blood pressure, AA that is released is preferentially converted to PGE1, PGE2, LXA4, resolvins, protectins and maresins that enhance renin synthesis whereas an increase in pressure and stretch stimuli that occurs secondary to an increase blood volume and blood pressure the released AA is converted to HEPETE and HETE that inhibit renin synthesis and release. Thus, AA (like PGE2 as discussed above) can have both stimulatory and inhibitory action on renin synthesis and release.

## 9. Renin Secretion at the Cellular Level

Dysregulation of renal renin synthesis and secretion can cause significant alterations in blood volume, blood pressure, and as a result significant damage to various organs could occur. Hence, understanding the control of renal renin synthesis and renal renin secretion is essential. In general, experiments are performed in experimental animals to study the control of renin at the organ and cellular level and in vitro systems. Although such studies do give valuable information about renin synthesis and secretion, one need to know that there could occur substantial differences between experimental animals and humans pertaining to quantitative differences in the number of renin-producing cells, the rate of intracellular renin processing, and the rate of renin secretion. In the adult kidney, renin is synthesized by JGA granular cells located close to the renal afferent arterioles at the entrance into the glomerular capillary network (see Figure 1). The transcription rate of the renin gene is an essential event that determines the production rate of renin.

Among the factors that regulate renin gene expression include cAMP and Ca^2+^pathways, cAMP-binding protein (CREB), and PPAR-γ (peroxisome proliferator-activated receptor-γ) that have a positive effect on renin gene transcription. On the other hand, NF-κB, IL-6, and TNF-α reduce renin gene expression and its secretion (see Figure 2, [34,35]. In addition, AA, and several of its metabolites (similarly EPA and DHA and their metabolites) also regulate renin gene expression [3,6,8,9,11,29]. In this context, it is noteworthy that AA (and EPA and DHA) can interact with syntaxin to regulate exocytosis and thus, control renin secretion.

Cyclic AMP stimulates renin secretion (by activating protein kinase A) [36]. Thus, catecholamines that activate cAMP formation (via β1-receptors) and PGI2 and PGE2 that inhibit cAMP degradation and nitric oxide (via cyclic GMP (cGMP)) and pharmacological inhibitors of cAMP-phosphodiesterases such as theophylline stimulate renin secretion [37]. In contrast to this, maneuvers that increase the cytosolic calcium concentration in renin-secreting cells inhibit renin release [38]. Thus, angiotensin-II and endothelin inhibit renin release [39,40]. Similarly, increased perfusion pressure of the afferent arterioles increases the calcium concentration in JGA cells and thus, inhibits renin release [41]. cGMP generation by ANP in renin-secreting cells also inhibits renin secretion [42].

Several neurotransmitters and neuropeptides stimulate renin secretion by enhancing cAMP pathway, whereas neuropeptide Y suppresses renin secretion by decreasing cAMP formation. Norepinephrine binds to β1-receptors on renin-secreting cells and thus, stimulates renin secretion. Renal vasoconstriction precedes sympathetic output to the kidneys [43,44,45,46,47,48,49,50,51,52]. The tubular macula densa cells and the preglomerular endothelial cells produce NO and prostaglandins (PGs) [53]. NO, PGI2 and PGE2 stimulate renin secretion [54,55].

## 10. AA Interacts with Syntaxin to Regulate Exocytosis/Renin Secretion

Renin is stored in granules in juxtaglomerular (JG) cells. SNAREs (soluble N-ethylmaleimide-sensitive factor attachment proteins), VAMP2 (vesicle associated membrane protein 2) and VAMP3 have a role in cAMP-stimulated exocytosis as is seen in other endocrine cells. VAMP2 and VAMP3 mRNA are expressed in JG cells and VAMP2 (but not VAMP3) is co-localized with renin-containing granules. Silencing VAMP2 blocks cAMP-induced renin release (by ~50%) whereas silencing VAMP3 had no effect [56]. This soluble NSF (N-ethylmaleimide-sensitive factor) attachment protein receptor (SNARE) protein VAMP2 mediated cAMP-stimulated renin release and exocytosis. Furthermore, JG cells with renin-containing granules co express and colocalize the isoform SNAP23 indicating that the SNARE protein SNAP23 is involved in cAMP-stimulated renin release, implying that renin release is a SNARE-dependent process [56,57,58,59].

Synaptic vesicle exocytosis is essential for neurotransmitter release at nerve terminals and renin by renin (JGA) cells. The vesicular transport machinery consists of four key components: (i) v-SNARE protein, which is a vesicle membrane protein, (ii) t-SNARE, the target membrane protein, (iii) membrane fusion *N*-ethylmaleimide-sensitive fusion cytosolic protein (NSF), and (iv) SNAPs that are adaptors for NSF termed (soluble NSF attachment proteins) (Figure 7). The v- and t-SNAREs are complementary to each other that are needed for vesicle docking. The assembled v- and t-SNARE acts as a receptor for the SNAPs that can incorporate the fusion protein, NSF. The docking and fusion particle containing all the four basic parts, thus formed, is called the SNARE complex. The energy needed for the vesicle fusion is derived from the hydrolysis of ATP by NSF, which is an ATPase [58]. Renin granule exocytosis needs SNAREs for the release of renin from juxtaglomerular cells [56,57,60]. Recent studies revealed that there is a critical role for AA and DHA in exocytosis of neurotransmitters in which the fusion of neurotransmitter-containing vesicles with the neuronal plasma membrane occurs. AA and DHA potentiate secretion from neurosecretory cells. AA increases soluble NSF attachment protein receptor complex and SNARE complex formation [61].

Syntaxin is one of the members of the SNARE family of proteins that mediates membrane fusion, an event needed for intracellular membrane trafficking [62]. Syntaxins diffuse in the plasma membrane to form clusters that have an important role in vesicle secretion process [63]. Exocytosis of secretory vesicles needs syntaxin to form clusters, which depends on the changes in the lipid bilayer of the cell membrane, that causes the aggregation of membrane proteins [64]. All SNARE proteins target membranes during fusion that is needed for exocytosis and is called the SNARE core complex, which is formed from SNARE proteins on the secretory vesicles (vesicle-associated membrane protein or VAMP) and a soluble SNARE protein (synaptosome-associated protein or SNAP). Syntaxins and VAMPs are anchored to the membranes (see Figure 7). Syntaxin 1 is needed for exocytosis of neurotransmitters. Membrane extension and neurotransmitter release needs fusion of intracellular vesicles with the plasma membrane involving SNARE protein assembly, membrane fusion, and its disassembly [65,66,67,68,69]. There are about 15 members of the syntaxin family. Syntaxins in conjunction with the cytoplasmic NSF and SNAP proteins mediate vesicle fusion and thus, are needed for exocytosis and endocytosis. All mammalian syntaxins, except for syntaxin 11, are transmembrane proteins and endosomal syntaxins is involved in specialized processes such as neurite outgrowth and myelin sheath formation [70].

It is evident from the preceding discussion that membrane-associated SNAREs, syntaxin 1, SNAP25 (synaptosomal-associated protein 25 kDa) and synaptobrevin are needed for membrane fusion [71]. Fusion of intracellular vesicular storage material with the cytosolic surface of the plasma membrane involving SNARE proteins is essential in membrane extension and exocytosis of neurotransmitters. The pairing of SNARE with syntaxin-1 and SNAP-25 on the intracellular surface of the plasma membrane is needed for neurotransmitter exocytosis. Phospholipases (PLs) that release AA are highly enriched in nerve growth cones and are involved in neurite outgrowth. Syntaxin 3 (STX3) that is needed for neurite growth is activated by AA and DHA suggesting that these fatty acids are essential for membrane expansion at the growth cones, and exocytosis of neurotransmitters [61,71,72,73,74,75,76]. Thus, AA and DHA are critical in the exocytosis of various neurotransmitters.

In a similar fashion, AA may modulate renin release from JGA. JG cells express VAMP2 and VAMP3 mRNA especially, VAMP2 that is explicitly co-localized with JGA granules that contain renin. VAMP2 plays a role in cAMP mediated renin exocytosis [56]. The expression and colocalization of SNAP23 in renin-containing granules of JG cells implies that SNAP23 is involved in renin release and thus, renin release is a SNARE-dependent process [56,57,58,59]. SNARE, syntaxin-1 and SNAP-25 interact with each other and are needed for exocytosis. Since both AA and DHA (AA > DHA) activate syntaxin, it is proposed that these fatty acids regulate renin release. Furthermore, several AA metabolites can either increase or decrease renin release [6,7,8,9,10,11,12,13,14], depending on the metabolite formed from AA. For instance, 12(s)-HPETE and 12-HETE and 12- and 15-LOX metabolites inhibit renin release whereas PGE2 and PGI2 increase renin release depending on their concentration and binding of PGE2 to EP1-EP4 receptors. Thus, AA can either increase or decrease renin release depending on the metabolite formed and its concentration and binding to specific receptors [6,7,8,9,10,11,12,13,14]. These results agree with the results that nucleus serves as a sensor and responds to changes in pressure and stretch {1–2 in the Figure 3} that results in the release of calcium, cPLA2 activation and release of AA [3] (and other unsaturated fatty acids from the membrane) that ultimately causes actomyosin force generation (AA can directly modulate cytoskeletal structures). This is akin to changes in the blood flow and vasomotor tone of the afferent arteriole of the glomerulus that is sensed by the cells of the JGA leading to changes in their nuclear membrane tension which results in the released AA (and of EPA and DHA) and their subsequent conversion to its products such as prostaglandins, thromboxanes, leukotrienes, lipoxins, and resolvins from EPA and DHA and protectins and maresins from DHA that can either increase or decrease blood flow and produce respective changes in the vasomotor tone. These changes ultimately result either an increase or decrease in renin release as the situation demands. Based on these evidences, it is envisaged that AA serves as a mechanotransducer of renin cell baroreceptor and thus, modulate renal blood flow, body fluid volume and blood pressure (see Figure 1, Figure 2, Figure 3, Figure 4 and Figure 8).

## 11. Conclusions and Therapeutic Implications

Exocytosis is the fusion of secretory vesicles with the plasma membrane that is essential for the discharge of vesicle content into the extracellular space. This is associated with incorporation of new proteins and lipids into the plasma membrane. Exocytosis can be seen in many cells {called as (constitutive exocytosis) or it can occur in specialized cells such as neurons, endocrine and exocrine cells (this includes insulin secretion by pancreatic β cells, renin by renin cells of JGA), called as regulated exocytosis}. Constitutive exocytosis is needed to secrete digestive proenzymes, peptide hormones, immunoglobulins, chylomicra lipoproteins by non-neuronal cells. In majority of the instances, exocytosis is triggered by an increase in the cytosolic free Ca^2+^ concentration. In neurons and endocrine cells, secretory vesicles fuse with the plasma membrane in response to stimulation, and their secretion is linked to synapsins (in neurons) or actin (in endocrine cells). Several Ca^2+^-binding proteins and synaptotagmin are essential for exocytosis at synapses. GTP-binding proteins are also involved in exocytosis. It is evident from the preceding discussion that SNARE that vesicle SNAREs (synaptobrevin and homologues) bind to target SNAREs (syntaxin/SNAP-25 and homologues), whereupon SNAPs and NSF bind to elicit membrane fusion and subsequent exocytosis [77]. Spatial control of exocytosis is needed for any biological processes such as neurotransmitter secretion, immune surveillance, cell migration and wound healing, and for development of cell polarity and cell growth (see Figure 9). In neurons, exocytosis of secretory granules and vesicles is confined to the synaptic cleft [78]. During immune surveillance, contact between an antigen-presenting cell (APC) and a cytotoxic T cell is essential and leads to formation of the immunological synapse that results in vesicle transport of soluble agents and membrane proteins within this zone of cell–cell contact that is needed to secrete cytokines, perforin, etc., to regulate immune response and induce apoptosis of tumor cells [79]. Proper cell migration and proliferation is needed for wound healing wherein exocytosis is coupled to local changes in plasma membrane and cytoskeleton dynamics [80]. During cell division, localized exocytosis occurs to mediate ingression of the cleavage furrow [81,82]. Development of cell polarity directs exocytosis at specialized sites of membrane growth [83,84].

In this context, it is important to note that AA and DHA and their metabolites participate and regulate neurite growth and neurotransmitters exocytosis, insulin and renin secretion, immune surveillance, cell migration, wound healing and cell growth and multiplication [6,7,8,9,10,11,12,13,14,61,71,72,73,74,75,76,85,86,87,88,89,90,91,92,93,94,95,96,97,98,99]. These results coupled with the observation that alterations in the pressure or stretch leads to an alteration in the nuclear membrane tension that triggers calcium release, cPLA2 activation and release of AA [4,5] implies the role of AA to function as a mechanotransducer. These results suggest that AA can function as a mechanotransducer. Released AA induces changes in the actin cytoskeleton behavior and other cytoskeleton structures (including lamin, integrin β1). This ultimately results in changes in the expression of genes and their products. The released AA can be metabolized to form a wide variety of its products based on the activities of COX, LOX and p450 enzymes that have potent biological actions (see Figure 5). These metabolites of AA have multiple actions that impinge on almost all types of cells and tissues and several physiological processes [100,101]. It is noteworthy that AA is an important component of all cell membranes. Hence, it is likely that all types of cells in the body that are exposed to pressure or stretch and/or have constitutive or regulated exocytotic function need AA for their physiological function. The pressure or stretch can be exerted by blood flow, changes in blood volume, blood viscosity and could be brought about by internal or external stimuli. It is likely that insulin secretion by pancreatic β cells, renin secretion by renin cells of JGA and vascular endothelial cells that are constantly exposed to shear stress of blood volume and changes in blood pressure are the most important cells/tissues that need constant supply of AA to bring about their respective physiological actions. Thus, it is suggested that secretion of insulin by β cells and renin by renin cells of JGA and vasodilator and vasoconstrictor molecules production by endothelial cells is regulated by their AA content and metabolism. This is supported by the reports that pancreatic β cell secretion of insulin and renin production are regulated by AA and its metabolites [6,7,8,9,10,11,12,13,14,86,87,88]. Vascular endothelial cells produce several vasoactive substances that regulate vasomotor tone and blood pressure [102,103,104]. Of all, AA metabolites PGI2, PGE2, LTs, TXs, LXA4, and dihydroxy fatty acids are critical (see Figure 5). The balance between vasoconstrictors and vasodilators produced from AA (and EPA and DHA) seem to have both local and systemic actions in the regulation of blood pressure. In this context, AA seems to be important since it forms the precursor to both vasoconstrictor (LTs, TXs) and vasodilator (PGI2, PGE2, and LXA4) metabolites. Furthermore, AA can regulate nitric oxide (NO) generation that is necessary to maintain vascular endothelial integrity, tissue perfusion and blood pressure [105,106,107,108]. The crosstalk between PGE2 (a vasodilator, platelet aggregator and pro-inflammatory molecule) and LXA4 (a vasodilator, platelet anti-aggregator and anti-inflammatory molecule) seems to play a critical role in insulin and renin secretion (see Figure 6 and Figure 9, [29,30,31]).

Since AA regulates exocytosis, it is imperative that cells (especially β, renin cells of JGA and vascular endothelial cells) need a constant supply of adequate amounts of AA. Since the availability of AA from diet is limited, cells/tissues depend on endogenous production of AA from dietary LA. Conversion of LA to AA needs desaturases and elongases enzymes (see Figure 5A), whereas the conversion of AA to their respective metabolites needs COX-1, COX-2, and 5-, 12-, and 15-LOX enzymes (see Figure 5). Based on the local conditions, AA and DHA are utilized to form their respective metabolites that determine the cellular process (such as exocytosis of insulin, renin, neurite growth, neurotransmitter release, immune response, cell growth, cell motility, wound healing, cancer cell growth or death or metastasis, etc.). Based on the available evidence, it is proposed that AA not only functions as a mechanotransducer to convey messages from the membrane to the nucleus but also appears to regulate cytoskeleton organization, cell shape, size, mitosis (meiosis) and gene expression [80,109,110,111,112,113,114,115]. 

It is evident from the current evidence that lipids are the prime constituents of the fusing membranes and play a critical role in exocytosis. In addition to fatty acids (AA and DHA) there appears to be a role for other lipids such as phosphatidylinositol-4,5-bisphosphate [PtdIns(4,5)P2], cholesterol, phosphatidic acid (PA), phosphatidylserine, phosphoinositides {such as PtdIns(4,5)P2, PtdIns(3)P, PIP2} also play a role in exocytosis [116]. It is noteworthy that AA forms an important constituent of these lipids (both structurally and functionally). Thus, the role of AA (and possibly, DHA and other unsaturated fatty acids) in exocytosis, immune surveillance, cell migration, wound healing, development of cell polarity and cell growth is definite. This is supported by the observation that plasma concentrations of AA in the phospholipid fraction are low in diabetes mellitus, hypertension, coronary heart disease, sepsis, pneumonia, rheumatoid arthritis, lupus (in RA and lupus decrease of GLA, the precursor of AA is also seen) (see Table 1 and Table 2, [117,118] diseases in which exocytosis, immune surveillance, cell migration, wound healing, cell growth are critical to recover. Atherosclerosis free aortae have abundant essential fatty acids (LA and ALA and AA is the metabolite of LA) whereas atherosclerotic lesions are deficient in EFAs. This suggests that vascular endothelial cell integrity and function is dependent on the availability of EFAs [119,120,121,122].

It is noteworthy that tumor cells are deficient in AA (see Table 3) and selectively undergo apoptosis (compared to normal cells) on supplementation with this fatty acid (reviewed in [3,123]. These results suggest that supplementation of adequate amounts of AA could be employed to selectively eliminate tumor cells. Similar approach can be implemented for the prevention and management of diabetes mellitus, hypertension, coronary heart disease, sepsis, pneumonia, rheumatoid arthritis, lupus, and other diseases. It is interesting that AA supplementation, when its deficiency exists, leads to no change or decrease in the formation of PGE2 but augments formation of anti-inflammatory, vasodilator, platelet anti-aggregator, and anti-cancer molecule LXA4 [124,125,126]. Although in the present discussion, emphasis has been on AA, it is likely that other unsaturated fatty acids such as LA, GLA, DGLA, ALA, EPA, DPA and DHA may show mechanotransducer function as well. Both our in vitro and in vivo studies revealed that of all the saturated and unsaturated fatty acids correct is the best in elaborating its cytoprotective, anti-inflammatory and anti-diabetic actions. These beneficial actions of AA are due to enhanced formation of LXA4, a potent anti-inflammatory and anti-cancer molecule. GLA, DGLA, EPA and DHA also enhanced LXA4 formation, but the amount formed is significantly less [33]. The ability of GLA, DGLA, EPA and DHA to enhance LXA4 formation could be attributed to their ability to displace AA from the cell membrane lipid pool. It was reported that resolvins and protectins enhance LXA4 formation and thus, mediate some, if not all, of their anti-inflammatory actions [6,82,127,128]. Our studies revealed that LXA4 is more potent than resolvins and protectins in its anti-inflammatory, cytoprotective and anti-cancer actions [129,130,131]). These results suggest that AA and LXA4 are more important and potent compared to other unsaturated fatty acids (such as GLA, DGLA, EPA and DHA) and their metabolites (resolvins, protectins and maresins) (reviewed in [3]). This may explain as to why studies performed with EPA and DHA did not result in dramatic response in the prevention of inflammation, diabetes mellitus and cancer due to deficiency of AA in these diseases and failure to its (AA) supplement in adequate amounts. Hence, it is suggested that AA need to be given along with EPA and DHA to obtain optimal results [119]. Similarly, methods need to be developed such that AA is selectively delivered to the JGA to regulate renin release and function. AA can be given orally, intravenously for long periods without significant side effects [3,132,133,134]. In view of these evidence, it is imperative that potential clinical use of a simple nutrient AA in several diseases need to be explored.

## Figures and Tables

**Figure 1 nutrients-14-00749-f001:**
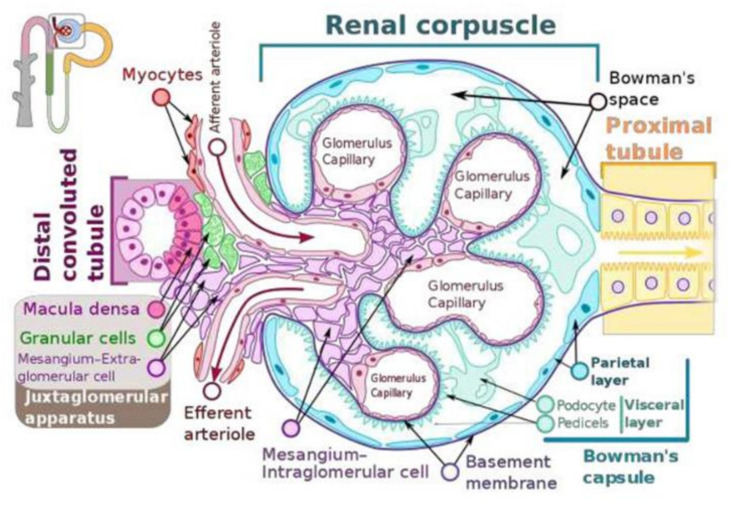
Cross section of the JGA and the glomerulus and their relationship to the afferent and efferent arterioles. Note the close relationship between JGA and the afferent and efferent arterioles. The JGA monitors the composition of the fluid in the distal convoluted tubule and adjusts the glomerular filtration rate accordingly. The picture shows the glomerulus and the surrounding structures. JGA = Juxtaglomerular apparatus.

**Figure 2 nutrients-14-00749-f002:**
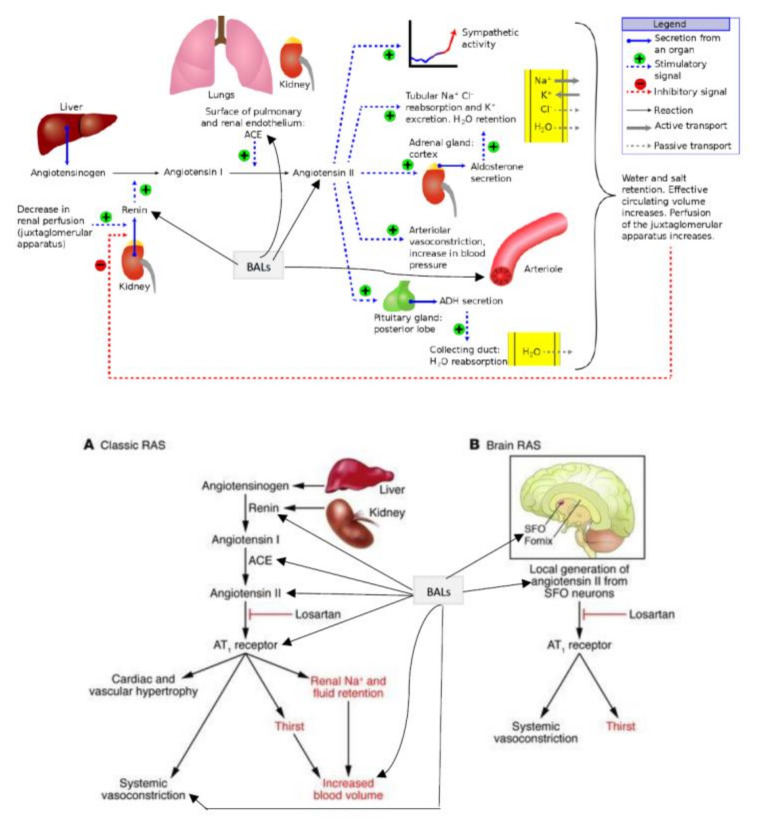
Scheme showing the release of renin from JGA, formation of angiotensin-II and regulation of fluid and electrolyte balance and blood pressure. (**A**) Classic RAS. Angiotensinogen from the liver is converted to angiotensin II by renin and angiotensin-converting enzyme (ACE). Angiotensin II activates AT1 receptors in target tissues producing changes in blood volume and blood pressure. (**B**) Brain RAS. Production of angiotensin II in the SFO affects thirst and salt appetite.IL-6 = Interleukin-6 TNF = Tumor necrosis factor RAS = Renin-angiotensin-aldosterone system AT receptor =Angiotensin receptor ACE = Angiotensin converting enzyme SFO = Subfornical organ.

**Figure 3 nutrients-14-00749-f003:**
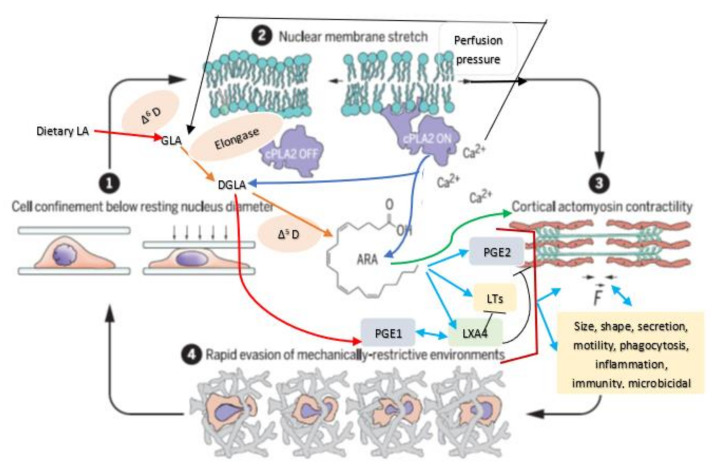
Scheme showing the mechanotransducer function of nucleus in response to changes in pressure and stretch. Pressure or stretch stimuli {1–2} increase nuclear membrane tension. This results in calcium release, cPLA2 activation and AA release {3} (other unsaturated fatty acids may also be released from the cell membrane) that results in actomyosin force generation (AA may also act on other cytoskeletal structures). AA metabolites regulate inflammation, immune response, and possess antimicrobial actions. Modified from Ref. [4].

**Figure 4 nutrients-14-00749-f004:**
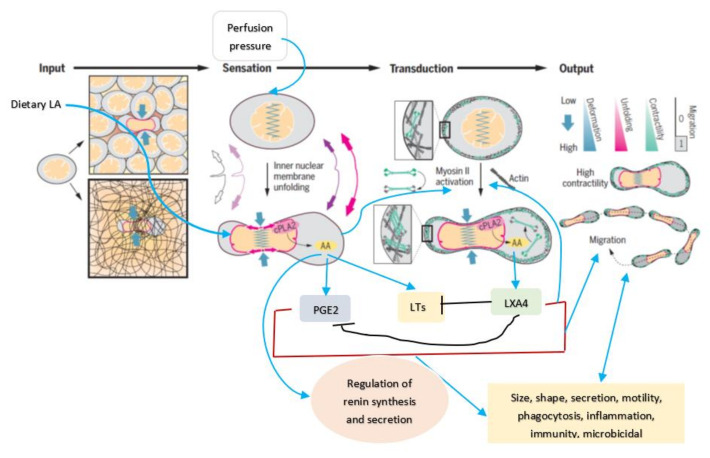
Scheme showing how nucleus can act as an elastic mechanotransducer in response to changes in the pressure and stretch stimuli (modified from Ref. [5]).

**Figure 6 nutrients-14-00749-f006:**
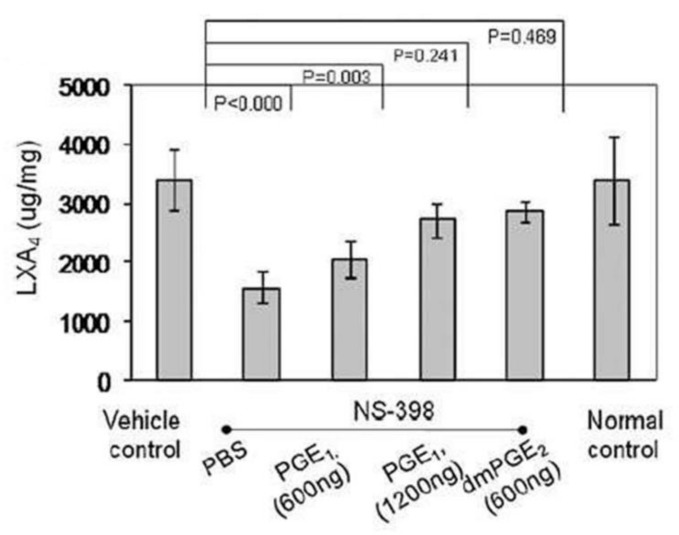
Regulation of LXA4 generation by PGE1 and PGE2.In experimental animals induced to develop collagen-induced arthritis, PGE1 and PGE2 enhance LXA4 formation PGE2 > PGE1). This data is taken from Ref. [31]. Normal control refers to the control foot pad of the opposite side.

**Figure 7 nutrients-14-00749-f007:**
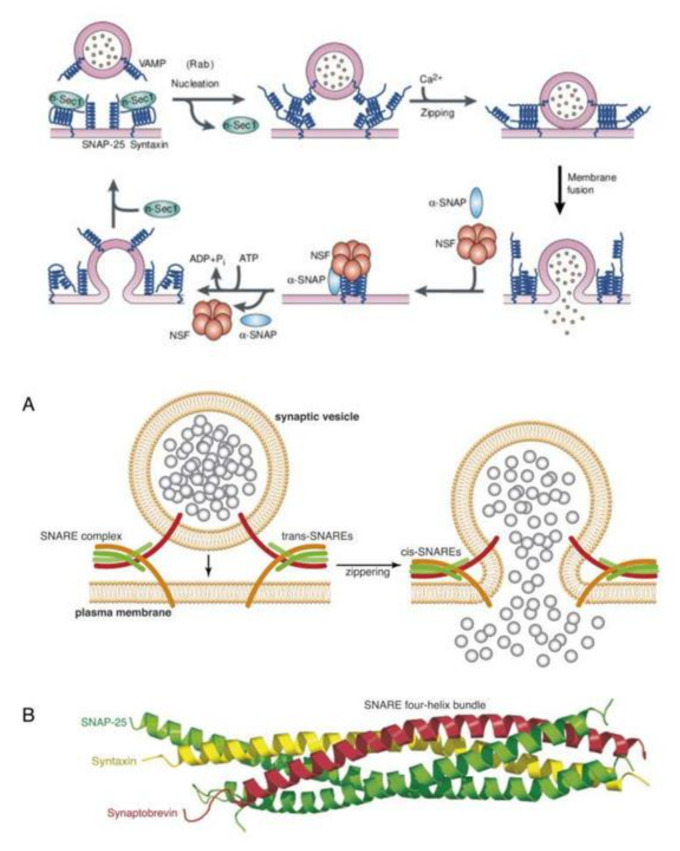
Schemes showing the mechanism of synaptic vesicle exocytosis. For details see text. (**A**) Scheme showing how SNAP-25 and Syntaxin participates in membrane fusion and exocytosis. (**B**) Scheme showing how SNAP-25, syntaxin and synaptobrevin interact with each other.

**Figure 8 nutrients-14-00749-f008:**
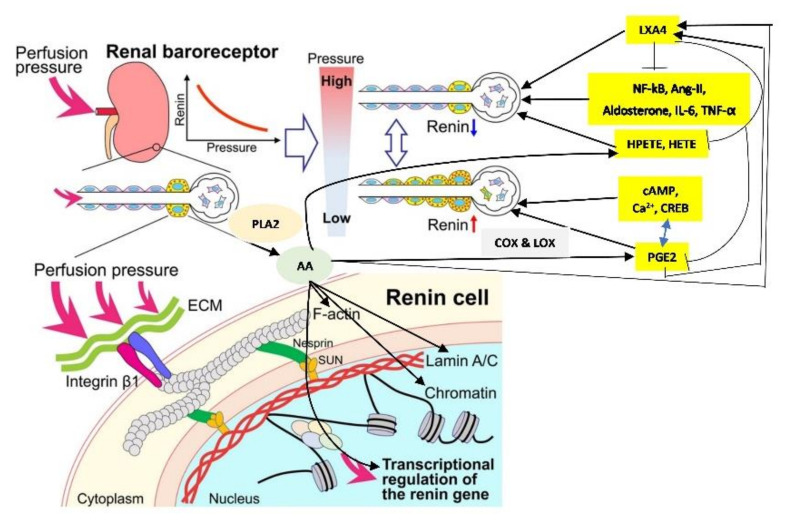
Scheme showing how AA can function as a mechanotransducer in response to changes in the perfusion pressure in the afferent arteriole and transmit renal baroreceptor actions. ECM = Extracellular matrix AA = Arachidonic acid. PGE2. +Prostaglandin E2 LXA4 = Lipoxin A4 NF-kB = Nuclear factor-kappa B IL-6 = Interleukin-6 TNF = Tumor necrosis factor HPETE = Hydroperoxyeicosatetraenoic acid HETE = Hydroxyeicosatetraenoic acid CREB = cAMP-binding protein. AA by itself or by its various metabolites can act on F-actin, Lamin A/C and chromatin to regulate renin gene expression and thus, regulate renin synthesis and exocytosis. Metabolites of AA such as PGE2, LXA4, HPETE and HETE can either enhance or decrease renin formation in response to changes in perfusion pressure (see Figure 8). PGE2 can either increase or decrease renin synthesis depending on its concentration and binding to its various receptors. PGE2, when it reaches its optimum level, triggers the synthesis of LXA4 from AA and LXA4, in turn, inhibits PGE2 formation. NF-kB, angiotensin-II, aldosterone, IL-6 and TNF-α stimulate PLA2 to induce the release of AA from cell membrane lipid pool and enhance the formation of PGE2 and other pro-inflammatory eicosanoids. LXA4 inhibits the expression of Nf-kB, and inhibits the synthesis of IL-6, TNF-α. HPETE and HETE are pro-inflammatory molecules formed from AA and can inhibit renin synthesis. This positive and negative feedback regulation among AA, PGE2, LXA4, NF-kB, IL-6, TNF-α, angiotensin-II and cAMP ensure that renin synthesis and exocytosis occur in an optimum way depending on the perfusion pressure, blood volume and blood pressure (see Figure 8).

**Figure 9 nutrients-14-00749-f009:**
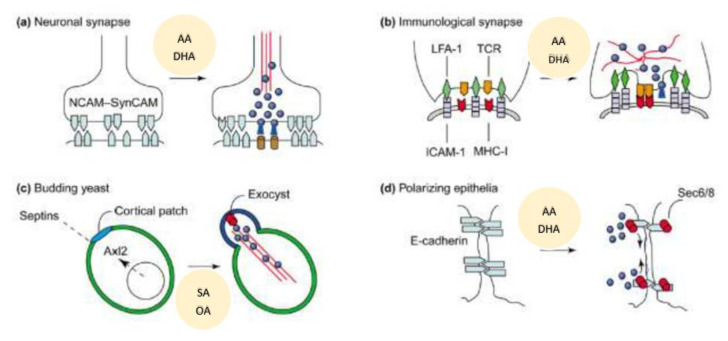
Molecular cues and membrane domains of localized exocytosis (modified from Ref. [83]). SA = Stearic acid. OA = Oleic acid. (**a**) Role of AA and DHA in neurite growth and neurotransmitters exocytosis {similar events lead to insulin and renin secretion}. (**b**) Under normal physiological conditions wherein robust immune surveillance exists, T-cell receptors (TCRs, orange) and LFA-1 cell adhesion molecules interact with MHC I and ICAM-1 molecules to form the immunological synapse. (**c**) In budding yeast, both extrinsic and intrinsic cues lead to cell–cell adhesion and is confined to E-cadherin. The exocyst is restricted to the tight junction as cells become more polarized. (**d**) SNARE proteins are localized at the site of delivery and fusion of intracellular vesicles because of local changes in intramembrane and membrane skeleton dynamics.

**Table 1 nutrients-14-00749-t001:** The percentage of distribution of fatty acids from plasma phospholipid fraction in patients with hypertension (HTN), coronary heart disease (CHD), type 2 diabetes mellitus, and diabetic nephropathy that are common with advanced age.

Fatty Acid	Control	HTN	CHD	Type	2 DM	Diabetic Nephropathy
16:0	25.9 ± 3.0	29.3 ± 2.7 *	27.8 ± 3.5	26.6	± 5.2	26.8 ± 2.7
18:0	20.9 ± 3.6	23.2 ± 4.9 *	18:0 ± 10.7	14.6	± 4.1	11.6 ± 3.6 *
18:1 n-9	13.0 ± 2.3	12.1 ± 1.5	11.5 ± 3.1	12.0	± 2.6	14.5 ± 3.1
18:2 n-6 (LA)	18.6 ± 3.1	14.5 ± 3.1 *	17.8 ± 5.0	13.9	± 5.3	15.1 ± 3.1
18:3 n-6 (GLA)	0.14 ± 0.1	0.4 ± 0.3 *	0.1 ± 0.1 *	0.2 ± 0.3		0.1 ± 0.2
20:3 n-6 (DGLA)	3.4 ± 1.0	3.1 ± 0.9	2.7 ± 1.1	1.7 ± 1.0 *		2.0 ± 0.8 *
20:4 n-6 (AA)	9.4 ± 1.8	7.8 ± 2.0 *	7.0 ± 2.1 *	4.6 ± 1.8 *		6.6 ± 2.6 *
22:5 n-6	0.7 ± 0.4	0.4 ± 0.4 *	1.0 ± 0.9	2.1 ± 0.6 *		1.3 ± 0.5 *
18:3 n-6/18:2 n-6	0.008	0.026	0.005	0.017		0.008
20:4 n-6/18:2 n-6	0.51	0.54	0.39	0.33		0.43
20:4 n-6/20:3 -6	2.8	2.53	2,59	2.8		3.3
18:3 n-3 (ALA)	0.2 ± 0.1	0.4 ± 0.2 *	0.3 ± 0.5	0.1 ± 0.2 *		0.1 ± 0.1 *
20:5 n-3 (EPA)	0.4 ± 0.4	0.6 ± 0.6	0.1 ± 0.2 *	0.3 ± 0.3		0.2 ± 0.3
22:5 n-3	0.5 ± 0.2	0.4 ± 0.5	0.3 ± 0.3 *	1.6 ± 1.3		1.7 ± 1.1
22:6 n-3 (DHA)	1.4 ± 0.5	1.2 ± 0.6	0.8 ± 0.4 *	0.5 ± 0.4 *		0.5 ± 0.3 *
20:5 n-3/18:3 n-3	1.8	1.39	0.41	3.2		4.0

All values are expressed as mean ± SD. * *p* < 0.05 compared to control. This data is taken from Refs [3,118].

**Table 2 nutrients-14-00749-t002:** Fatty acid analysis of the plasma PL (phospholipid) fraction in patients with pneumonia, Figure 3 and [117].

	Control (n = 10)	Pneumonia (n = 12)	Septicemia (n = 14)	RA (n = 12)	SLE (Lupus) (n = 5)
16:0	24.8 ± 3.4	32.5 ± 3.6	26.95 ± 4.1	30.2 ± 3.0	32.0 ± 3.75
18:0	23.3 ± 4.1	21.4 ± 7.1	24.58 ± 6.0	19.0 ±6.1	14.6 ± 5.82
18:1 n-9	13.1 ± 2.3	15.6 ± 3.2	16.5 ± 3.3 *	14.8 ± 2.1	16.0 ± 2.78
18:2 n-6	17.7 ± 3.1	14.2 ± 0.3 *	16.3 ± 2.4	17.5 ± 2.7	20.8 ± 2.2
18:3 n-6	0.13 ± 0.09	0.13 ± 0.08	0.04 ± 0.05 *	0.02 ± 0.04 **	0.01 ± 0.01 **
20:3 n-6	3.2 ± 0.79	1.5 ± 0.4 *	0.46 ± 0.54 *	2.5 ± 0.58	2.12 ± 0.52
20:4 n-6	8.8± 2.0	5.1 ± 0.4 *	5.8 ± 1.6 *	9.5 ± 2.2	8.93 ± 2.0
22:4 n-6	0.42 ± 0.23	0.8 ± 0.9	0.34 ± 0.28	0.26 ± 0.37 **	0.18 ± 0.18 **
22:5 n-6	0.73 ± 0.55	0.45 ± 0.63	1.5 ± 1.02 *	0.6 ± 0.7	0.8 ± 1.0
18:3 n-3	0.27 ± 0.12	0.09 ± 0.04 *	0.16 ± 0.11 *	0.12 ± 0.16 *	0.1 ± 0.1 *
20:5 n-3	0.25 ± 0.26	0.23 ± 0.24	0.01 ± 0.01 *	0.05 ± 0.14 **	0.04 ± 0.04 **
22:6 n-3	1.43 ± 0.43	0.54 ± 0.43 *	1.2 ± 1.14	0.62 ± 0.56 *	0.88 ± 0.75 *

** *p* < 0.001 compared to control; * *p* < 0.05 compared to control.

**Table 3 nutrients-14-00749-t003:** Content of fatty acids in normal liver, hepatoma cells, and in microsomal suspensions from normal liver and Yoshida hepatoma cells. All values of mean ± S. E. This data is form Ref. [3].

Measurement (Fatty Acid)	Normal Intact Liver	Intact Yoshida Cells	Normal Liver Microsomes	Yoshida Microsomes
16:0	18.5 ± 0.2	18.7 ± 2.0	18.9 ± 1.1	18.5 ±0.5
18:0	17.5 ± 0.5	13.3 ± 1.1	22.0 ± 3.0	13.7 ± 0.2
18:1, n-9 (oleic acid)	12.1 ± 1.0	21.5 ± 0.8	8.6 ± 1.0	18.1 ± 0.3
20:4 (AA)	16.7 ± 2.4	8.7 ± 0.7	19.1 ± 2.4	9.6 ± 0.8
22:5	-	2.9 ± 0.1	-	2.4 ± 0.3
22:6 (DHA)	6.3 ± 0.2	5.2 ± 0.6	6.1 ± 0.3	5.3 ± 0.4

## Data Availability

All the data has been given in the manuscript/references.

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
