# Peer review of "Arachidonic Acid as Mechanotransducer of Renin Cell Baroreceptor"

_nutrients, 2022, doi:10.3390/nu14040749_

Round 1

Reviewer 1 Report

The manuscript describe arachidonic acid and its metabolites adjust blood pressure via renin production in the juxtaglomerular cells.  I have one request to the author.

Conclusion should be shorten, it would better for readers.

Author Response

I have read the reviewers comment. Since the topic is new and novel, it has become necessary to conclude in a lengthy manner. I feel that this is necessary so that the topic is well explained to the readers. Hence bery minor editing of the conclusion section was done.

Reviewer 2 Report

1) This work represents a complex and rich state of the art of possible pathways for the participation of arachidonic acid (AA) and its metabolites in the modulation of renin and in the modifications of the cytoskeleton of juxtaglomerular cells.

2) Please, I would recommend the author to add a glossary for abbreviations just before the "References" section. Also, check that all the first abbreviations have their description.

3) In the same way, please, the author must indicate in the figures what these abbreviations are.

4) Figure 6. I do not know if the graph is redone with the data from reference number 30 or if the figure has been borrowed with express authorization. If not, please indicate only the differences in the text and delete the figure. It is also unclear whether "normal control" refers to the contralateral side without arthritis or to a positive control (LXA4 loading at high concentrations). If it were a healthy control, I do not understand well the values ​​of that graph. Please, I would appreciate a better description of that figure.

5) In tables 1 and 3 you should also indicate the references at the bottom of the table.

 6) "Conclusions" section. The author comments on AA as a clinical treatment in certain diseases, but a more specific and focused AA conclusion on AA as a treatment in granule cells and their renin-angiotensin system (RAS) is just missing. Please, I would appreciate if the author better specify that aspect that gives mention to the title of the manuscript.

Author Response

I thank the reviewer for his useful comments.

As sugegsted abbreviations have been given at the end of he text of the manuscript before the references. 

Similarly all the abbreviations have been given at the end of the figures also as suggested.

The reference from which data for Table 3 was taken was also given.

 Figure 6 is from reference 30. This has ben mentioned in the legend to the figure. In tis figure normal control refers to the healthy contralateral limb. This has been mentioned in the revised manuscript.

As suggested by the reviewer, a sentence has been added in the conclusion section that methods to deliver AA selectively to JGA need to be developed to regulate renin secretion. 

Round 2

Reviewer 2 Report

I appreciate the modifications made by the author.

Author Response

Suggested modifications have been incorporated in the revised manuscript.